# A short plus long-amplicon based sequencing approach improves genomic coverage and variant detection in the SARS-CoV-2 genome

**Carlos Arana[1,2], Chaoying Liang[1,2], Matthew Brock[1,2], Bo Zhang[1], Jinchun Zhou[1,2], Li Chen[3], Brandi Cantarel[4], Jeffrey SoRelle[3], Lora V. Hooper[1,5], Prithvi Raj**[1,2]*

**1** Department of Immunology, University of Texas Southwestern Medical Center, Dallas, TX, United States of America, **2** Microbiome and Genomics core, University of Texas Southwestern Medical Center, Dallas, TX, United States of America, **3** Department of Pathology, University of Texas Southwestern Medical Center, Dallas, TX, United States of America, **4** Department of Bioinformatics, University of Texas Southwestern Medical Center, Dallas, TX, United States of America, **5** Howard Hughes Medical Institute, University of Texas Southwestern Medical Center, Dallas, TX, United States of America

* prithvi.raj@utsouthwestern.edu

**Data Availability Statement:** Raw sequencing data have been deposited to the NCBI Sequence Read Archive (accession ID PRJNA729878).

## Abstract

High viral transmission in the COVID-19 pandemic has enabled SARS-CoV-2 to acquire new mutations that may impact genome sequencing methods. The ARTIC.v3 primer pool that amplifies short amplicons in a multiplex-PCR reaction is one of the most widely used methods for sequencing the SARS-CoV-2 genome. We observed that some genomic intervals are poorly captured with ARTIC primers. To improve the genomic coverage and variant detection across these intervals, we designed long amplicon primers and evaluated the performance of a short (ARTIC) plus long amplicon (MRL) sequencing approach. Sequencing assays were optimized on VR-1986D-ATCC RNA followed by sequencing of nasopharyngeal swab specimens from fifteen COVID-19 positive patients. ARTIC data covered 94.47% of the virus genome fraction in the positive control and patient samples. Variant analysis in the ARTIC data detected 217 mutations, including 209 single nucleotide variants (SNVs) and eight insertions & deletions. On the other hand, long-amplicon data detected 156 mutations, of which 80% were concordant with ARTIC data. Combined analysis of ARTIC + MRL data improved the genomic coverage to 97.03% and identified 214 high confidence mutations. The combined final set of 214 mutations included 203 SNVs, 8 deletions and 3 insertions. Analysis showed 26 SARS-CoV-2 lineage defining mutations including 4 known variants of concern K417N, E484K, N501Y, P618H in spike gene. Hybrid analysis identified 7 nonsynonymous and 5 synonymous mutations across the genome that were either ambiguous or not called in ARTIC data. For example, G172V mutation in the ORF3a protein and A2A mutation in Membrane protein were missed by the ARTIC assay. Thus, we show that while the short amplicon (ARTIC) assay provides good genomic coverage with high throughput, complementation of poorly captured intervals with long amplicon data can significantly improve SARS-CoV-2 genomic coverage and variant detection.

**Funding:** Present study was supported by Microbiome Research Laboratory at UT Southwestern Medical Center.

**Competing interests:** The authors declare no competing interests.

## Introduction

Severe acute respiratory syndrome coronavirus 2 (SARS-CoV-2), the causative pathogen for COVID-19 disease, continues to impact the global population with a growing number of variants [1]. Community spread is the predominant mechanism leading to the increasing incidence of COVID-19 disease world-wide. SARS-CoV-2 genome sequencing data suggest that several novel variants and regional strains are emerging in the United States [2, 3]. Whole genome sequencing (WGS) analysis of these viruses enables high resolution genotyping of circulating viruses to identify emerging strains [4, 5]. Genome sequencing is a powerful tool that can be used to understand the transmission dynamics of outbreaks and the evolution of the virus over time [6]. Phylogenetic analysis of WGS data can reveal a virus's origin and genetic diversity of circulating strains of the virus [7, 8]. With the ongoing pandemic, SARS-CoV-2 is getting ample opportunities to replicate and incorporate new mutations that can potentially impact virus characteristics such as transmissibility as reported in the cases of the B.1.1.7 lineage in England, B.1.351 lineage in South Africa and B.1.617.2 in India [9–12]. In addition, the abundance of mutations in new strains can also impact the performance of diagnostic and research methods that were developed based on the original reference genome from the beginning of the pandemic last year [13–15]. Therefore, methods and strategies of virus detection and genome sequencing need to be updated.

The ARTIC network protocol is one of the most widely used methods to sequence the SARS-CoV-2 genome [8, 16]. Two pools of primers in this assay amplify multiple short amplicons to assemble the entire genome. An increasing number of mutations in emerging strains poses one potential challenge to this strategy, as mutated primer binding sites may cause amplicon dropout or uneven sequencing coverage resulting in the loss of information or inaccurate data. To address this issue, we developed a new approach that amplifies each specimen using both, short and long amplicon primers to uniformly capture the entire viral genome. This approach offers two potential benefits. First, a smaller number of primers needed to amplify the entire genome reduces the chance of encountering a mutated site. Second, besides single nucleotide changes, deletions and insertions can be captured more effectively with longer amplicons and sequencing reads. Here, we present our approach to supplement ARTIC's short amplicon sequences with long-amplicons to generate more complete and high-quality sequencing data for mutation detection and phylogenetic analysis (Fig 1A).

## Materials and methods

### a. Sample

We used ATCC VR-1986D as our positive control RNA and a human dendritic cell RNA as our negative control to test our own primer design (MRL-Primer) and ARTIC primers. Next, we tested RNA from fifteen SARS-CoV-2 virus positive nasopharyngeal swabs received in universal transport media. Alinity M SARS-CoV-2 AMP Kit on an automated Alinity system was used for qPCR. Samples with Ct value <30 were used for investigation. Samples were de-identified and analyzed with a waiver from UT Southwestern Institutional Review Board. Study specimens were de-identified before authors have access to them. Research team had no access to any identifying information.

### b. Primers

We designed our own set of primers that amplify long-amplicons spanning 1.5–2.5Kb of the SARS-CoV-2 viral genome. Details on our primer sequences are provided in S1 File. ARTIC

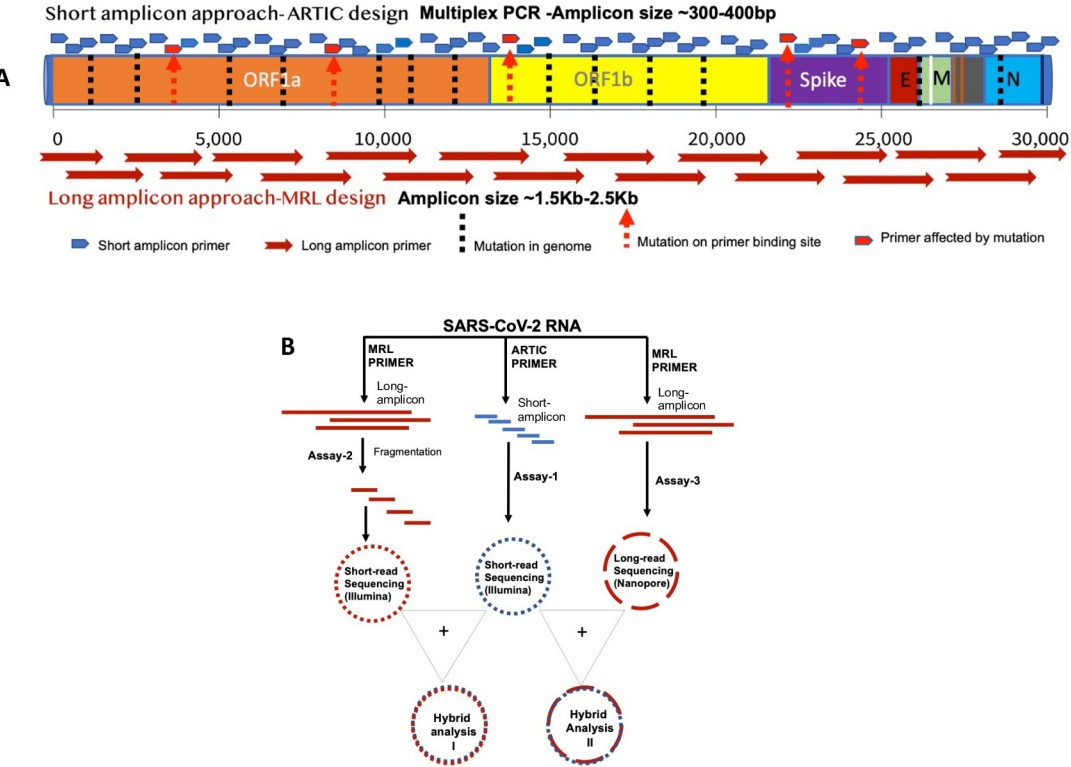

**Fig 1. Study rationale and assay design.** Panel A illustrates the study's rationale and sketches the layout of the ARTIC and MRL primer pools across the SARS-CoV-2 genome. Small and long arrows indicate short and long amplicons, respectively. Red arrows indicate a mutation with potential to alter the primer binding site. Panel B shows the design of the three assays developed and assessed in the present study. Assay-1 is based on short-amplicons generated with the ARTIC primer pool. Assay-2 is based on long-amplicons made with MRL primers followed by short-read sequencing on MiSeqDx. Assay-3 is based on long-amplicons sequenced by long read sequencing technology on MinION platform.

data was generated using ARTIC nCoV-2019 V3 Panel primers [16] purchased from Integrated DNA technologies (IDT).

## c. Assays and protocol

We developed three assays to sequence the SARS-CoV-2 genome and assessed their performance in the present study (Fig 1B). In Assay-1, virus genome was amplified using the ARTIC primer pool only and sequenced on the MiSeqDx Illumina platform. In second assay, we used our own primer design (MRL Primer) to generate 19 long amplicons of 1.5–2.5Kb size to capture the complete genome of the virus. These long amplicons were then fragmented into 300-500bp sizes and sequenced on the MiSeqDx platform. The third assay also used our own primers (MRL Primers), but the long-amplicons were directly sequenced on the long-read sequencing platform, MinION, from Oxford Nanopore Technology (ONT). Finally, the performance of the individual assays as well as the combined assays, (Hybrid 1 & Hybrid 2), were assessed.

**Step 1: RNA extraction and quality control.** RNA was extracted from nasopharyngeal swabs using the Chemagic Viral DNA/RNA 300 Kit H96 (Cat# CMG-1033-S) on a Chemagic 360 instrument (PerkinElmer, Inc.) following the manufacturer's protocol. A sample plate, elution plate, and a magnetic beads plate were prepared using an automated liquid handling instrument (Janus G3 Reformatter workstation, PerkinElmer Inc). In brief, an aliquot of 300μl from each sample, 4μL Poly(A) RNA, 10μL proteinase K and 300μL lysis buffer 1 were added

to respective wells of a 96 well plate. The sample plate, elution plate, (60μL elution buffer per well) and magnetic beads plate (150μL beads per well) were then placed on a Chemagic 360 instrument and RNA was extracted automatically with an elution volume of 60μL. No direct quantification of extracted RNA was done due to small concentration. Ct values of <30 was used as inclusion criterion.

**Step 2: cDNA synthesis and PCR amplification of SARS CoV-2 genome using gene-specific primers.** We used Invitrogen SuperScript™ III One-Step RT-PCR System with Platinum™Taq High Fidelity DNA Polymerase (Catalog Number: 12574–035) to make and amplify cDNA. SARS Cov-2 gene-specific primer set *MRL*-design and ARTIC design were synthesized from IDT. The MRL primer set included two primer pools (19 pairs, about 1.5Kb to 2.5Kb/ amplicon, Tm 59–60˚C), whereas ARTIC CoV-2019 primer pools included 109 pairs (about 400nt/ amplicon, Tm 60–62˚C). Both primers were used to amplify ATCC VR-1986D genomic RNA from severe acute respiratory syndrome-related coronavirus 2 Positive control. We also used a negative control of Human RNA extracted from monocyte-derived dendritic cells (MDDCs) to test our assay. We started with 0.1 ng of ATCC VR-1986D RNA (8000 genome copies), 100 ng of human MDDCs RNA, and 3–6 ul of Covid -19 patient RNA (RT-PCR Ct value 30) as an input RNA amount per each cDNA synthesis reaction. Reverse transcription was performed at 50˚C for 30 min, followed by denaturation at 94˚C for 2 min. PCR amplification involved 35 cycles (95˚C for 30 s, 55˚C for 1 min, 68˚C for 4.5 min) followed by a final extension at 68˚C for 10 min. Reaction products from two primer pools were combined and a bead-based cleanup was performed. Agencourt AMPure XP beads by Beckman Coulter (Catalog# A63881) were used for purification. Then, the cDNA quantity was measured using the Picogreen method. Quant-iT™ PicoGreen dsDNA Assay kit by Invitrogen with Catalog # P7589 and a PerkinElmer plate reader (PerkinElmer Victor X3, 2030 Multilabel Reader) were used in the assessment; and the cDNA quality was verified with a Bioanalyzer (Agilent High Sensitivity DNA kit, Catalog # 5067–4626).

**Step 3: NGS workflow- amplicon library preparation and quality control.** We used the Kapa HyperPlus Library Preparation Kit (Catalog #KK8514) to construct our sequencing libraries. The cDNA input amount for each library preparation was between 50–500 ng due to the limitation of cDNA quantity. We started with a 5-minute 37˚C enzymatic fragmentation for the cDNA amplicons generated by MRL-primer. Since ARTIC amplicon size w is already around 400bp, no fragmentation was performed on these replicates. Then end-repair, A-tailing, and UMI adapter ligation were performed on the amplicons. After the ligation step, we performed a double-sided size selection with AMPure XP beads followed by 4 to 8 PCR cycles to amplify adapter ligated fragments. Finally, the amplified libraries were purified by AMPure XP beads to form the final libraries. The libraries' quantity was measured with Picogreen, and the quality was verified with a Bioanalyzer (Agilent DNA 1000 kit, catalog # 5067–1504). In addition, we did qPCR to check the adapter ligation efficiency using Applied Biosystems 7500 Real-Time PCR instrument.

**Step 4: MiSeqDx sequencing.** About 20pM barcoded libraries were sequenced on MiSeqDx sequencer using 600 cycle v3 flow cell kit. About 5% PhiX DNA was added to the sequencing run to increase diversity. The summary of sequencing metrics is given in Table 1.

**Step 5: Nanopore library preparations and sequencing.** The leftover cDNA amplicons from step 2 were used to generate libraries for long-read sequencing analysis on MinION. EXP-NBD104 and SQK-LSK109 kits were used with Oxford Nanopore's (ONT) native barcoding protocol (Version: NBE_9065_v109_revV_14Aug2019) to construct the libraries. Due to limited sample concentration, this assay was only done on positive control ATCC RNA and 3 patient samples. Library preparation was performed according to the manufacturer's recommended protocol. Native barcoded libraries were pooled in equimolar concentrations and

**Table 1. Summary of sequencing metrics for all assays on fifteen samples.**

| Assay | Sample | Sample category | RT-PCR Ct-value | # Input reads | % Non-host reads (Kraken 2) | # Trimmed reads (fastp) | # Mapped reads | % Mapped reads | Coverage median | % Coverage > 1x | % Coverage > 10x | Genome Fraction | # SNPs (BCFTools) | Pangolin lineage (BCFTools) |
|---|---|---|---|---|---|---|---|---|---|---|---|---|---|---|
| ARTIC ASSAY-I | A10 | Patient | 5.47 | 2221344 | 99.83 | 1446798 | 1439538 | 99.5 | 2328 | 100 | 100 | 99.90% | 36 | B.1.1.7 |
| | A12 | Patient | 17.93 | 1932150 | 99.12 | 1242820 | 1220034 | 98.17 | 561 | 100 | 97 | 98.10% | 34 | B.1.1.7 |
| | A4 | Patient | 12.79 | 1871886 | 99.33 | 1233994 | 1213933 | 98.37 | 3416 | 100 | 100 | 99.90% | 38 | B.1.617.1 |
| | A5 | Patient | 14.43 | 2494742 | 94.26 | 1463822 | 1330372 | 90.88 | 629 | 100 | 95 | 95.20% | 37 | B.1.1.7 |
| | B10 | Patient | 24.7 | 4187664 | 43.65 | 2306338 | 356185 | 15.44 | 206 | 98 | 91 | 91.70% | 25 | B.1.617.1 |
| | B11 | Patient | 25 | 4800614 | 39.5 | 2599098 | 223929 | 8.62 | 111 | 97 | 86 | 86.30% | 22 | A |
| | B5 | Patient | 12.98 | 2730112 | 44.61 | 1535582 | 282773 | 18.41 | 208 | 98 | 90 | 90.10% | 34 | P.1.1 |
| | B9 | Patient | 12.93 | 2462484 | 96.54 | 1511158 | 1419156 | 93.91 | 761 | 99 | 96 | 96.80% | 33 | B.1.1.7 |
| | C5 | Patient | na | 2001802 | 96.13 | 1288162 | 1207630 | 93.75 | 894 | 99 | 97 | 97.60% | 27 | B.1.351 |
| | C6 | Patient | na | 2941214 | 65.69 | 1600492 | 689507 | 43.08 | 417 | 99 | 93 | 93.80% | 19 | B.1.526 |
| | ATCC | Control RNA | contrl | 525512 | 99.88 | 341022 | 338500 | 99.26 | 244 | 99 | 90 | 90.90% | 2 | A |
| | NP1 | Patient | <30 | 1560066 | 97.75 | 493404 | 483179 | 97.93 | 403 | 100 | 97 | 97.40% | 23 | B.1.2 |
| | NP2 | Patient | <30 | 2634944 | 99.97 | 997780 | 996868 | 99.91 | 2679 | 100 | 100 | 99.90% | 26 | B.1.2 |
| | NP3 | Patient | <30 | 1382532 | 99.96 | 473600 | 472764 | 99.82 | 759 | 100 | 99 | 99.10% | 26 | B.1.2 |
| | NP4 | Patient | <30 | 2678906 | 93.22 | 806390 | 763484 | 94.68 | 310 | 99 | 91 | 91.40% | 18 | B.1.575.1 |
| | NP5 | Patient | <30 | 5279922 | 45.44 | 935280 | 469650 | 50.21 | 135 | 98 | 88 | 88.80% | 14 | B.1.427 |
| MRL ASSAY-II | A10 | Patient | 5.47 | 2518188 | 99.64 | 2078572 | 2069729 | 99.57 | 689 | 98 | 77 | 76.90% | 29 | B.1.1.7 |
| | A12 | Patient | 17.93 | 2478842 | 95.13 | 1972268 | 1867067 | 94.67 | 42 | 89 | 68 | 68.40% | 25 | B.1.1.7 |
| | A4 | Patient | 12.79 | 2911838 | 98.97 | 2350034 | 2321911 | 98.8 | 811 | 98 | 79 | 78.70% | 37 | B.1.609 |
| | A5 | Patient | 14.43 | 3096338 | 64.66 | 2163730 | 1314659 | 60.76 | 75 | 84 | 69 | 68.40% | 23 | B.1.1.7 |
| | B10 | Patient | 24.7 | 2521036 | 15.02 | 1574540 | 69188 | 4.39 | 3 | 67 | 32 | 31.50% | 13 | None |
| | B11 | Patient | 25 | 3129308 | 15.17 | 1893714 | 62517 | 3.3 | 1 | 53 | 18 | 17.20% | 6 | None |
| | B5 | Patient | 12.98 | 2848068 | 15.36 | 1799694 | 95309 | 5.3 | 4 | 65 | 33 | 33.00% | 10 | None |
| | B9 | Patient | 12.93 | 3235630 | 77.65 | 2455908 | 1871924 | 76.22 | 32 | 90 | 70 | 69.60% | 25 | B.1.1.7 |
| | C5 | Patient | na | 2422548 | 80.11 | 1882074 | 1487868 | 79.05 | 255 | 98 | 80 | 81.30% | 21 | B.1.351 |
| | C6 | Patient | na | 3275182 | 24.02 | 2046190 | 312684 | 15.28 | 4 | 67 | 36 | 35.30% | 8 | None |
| | ATCC | Control RNA | contrl | 700414 | 99.47 | 566400 | 562121 | 99.24 | 10 | 78 | 50 | 79.80% | 5 | A |
| | NP1 | Patient | <30 | 5909444 | 97.82 | 3243578 | 3190145 | 98.35 | 105 | 99 | 82 | 81.90% | 25 | B.1 |
| | NP3 | Patient | <30 | 7250976 | 99.98 | 2540408 | 2538292 | 99.92 | 637 | 98 | 98 | 98.40% | 26 | B.1.2 |
| | NP4 | Patient | <30 | 6073616 | 79.48 | 1876650 | 1558882 | 83.07 | 54 | 96 | 79 | 79.30% | 17 | B.1 |
| | NP5 | Patient | <30 | 4074880 | 10.75 | 750598 | 120598 | 16.07 | 6 | 83 | 38 | 38.80% | 9 | None |

*(Continued)*

**Table 1.** (Continued)

| | Sample | Sample category | RT-PCR Ct-value | # Input reads | % Non-host reads (Kraken 2) | # Trimmed reads (fastp) | # Mapped reads | % Mapped reads | Coverage median | % Coverage > 1x | % Coverage > 10x | Genome Fraction | # SNPs (BCFTools) | Pangolin lineage (BCFTools) |
|---|---|---|---|---|---|---|---|---|---|---|---|---|---|---|
| **HYBRID** (ARTIC +MRL) | A10 | Patient | 5.47 | 4739532 | 99.72 | 3525370 | 3509267 | 99.54 | 4817 | 100 | 100 | 99.90% | 37 | B.1.1.7 |
| | A12 | Patient | 17.93 | 4410992 | 96.67 | 3215088 | 3087101 | 96.02 | 1088 | 100 | 99 | 99.90% | 34 | B.1.1.7 |
| | A4 | Patient | 12.79 | 4783724 | 99.09 | 3584028 | 3535844 | 98.66 | 7053 | 100 | 100 | 99.90% | 39 | B.1.617.1 |
| | A5 | Patient | 14.43 | 5591080 | 76.61 | 3627552 | 2645031 | 72.92 | 1110 | 100 | 98 | 97.50% | 38 | B.1.1.7 |
| | B10 | Patient | 24.7 | 6708700 | 32.03 | 3880878 | 425373 | 10.96 | 265 | 98 | 94 | 94.70% | 26 | B.1.617.1 |
| | B11 | Patient | 25 | 7929922 | 29.24 | 4492812 | 286446 | 6.38 | 142 | 99 | 88 | 88.20% | 24 | A |
| | B5 | Patient | 12.98 | 5578180 | 28.82 | 3335276 | 378082 | 11.34 | 262 | 99 | 94 | 94.10% | 36 | P.1.1 |
| | B9 | Patient | 12.93 | 5698114 | 84.84 | 3967066 | 3291080 | 82.96 | 1256 | 100 | 98 | 98.60% | 34 | B.1.1.7 |
| | C5 | Patient | na | 4424350 | 86.62 | 3170236 | 2695498 | 85.03 | 1868 | 100 | 99 | 98.70% | 27 | B.1.351 |
| | C6 | Patient | na | 6216396 | 42.31 | 3646682 | 1002191 | 67.48 | 491 | 99 | 95 | 94.90% | 19 | B.1.526 |
| | ATCC | Control RNA | contrl | 12412298 | 99.97 | 4124970 | 4112354 | 99.69 | 1505 | 100 | 100 | 99.90% | 3 | A |
| | NP1 | Patient | <30 | 7469510 | 97.99 | 2399616 | 2359417 | 98.32 | 578 | 100 | 99 | 99.10% | 25 | B.1.596 |
| | NP3 | Patient | <30 | 8633508 | 99.98 | 2669512 | 2666401 | 99.88 | 3470 | 100 | 100 | 99.90% | 26 | B.1.2 |
| | NP4 | Patient | <30 | 8752522 | 84.57 | 2439048 | 2136213 | 87.58 | 509 | 100 | 98 | 98.00% | 19 | B.1.575 |
| | NP5 | Patient | <30 | 9354802 | 29.99 | 1685878 | 590248 | 75.01 | 195 | 99 | 92 | 92.10% | 16 | B.1.427 |

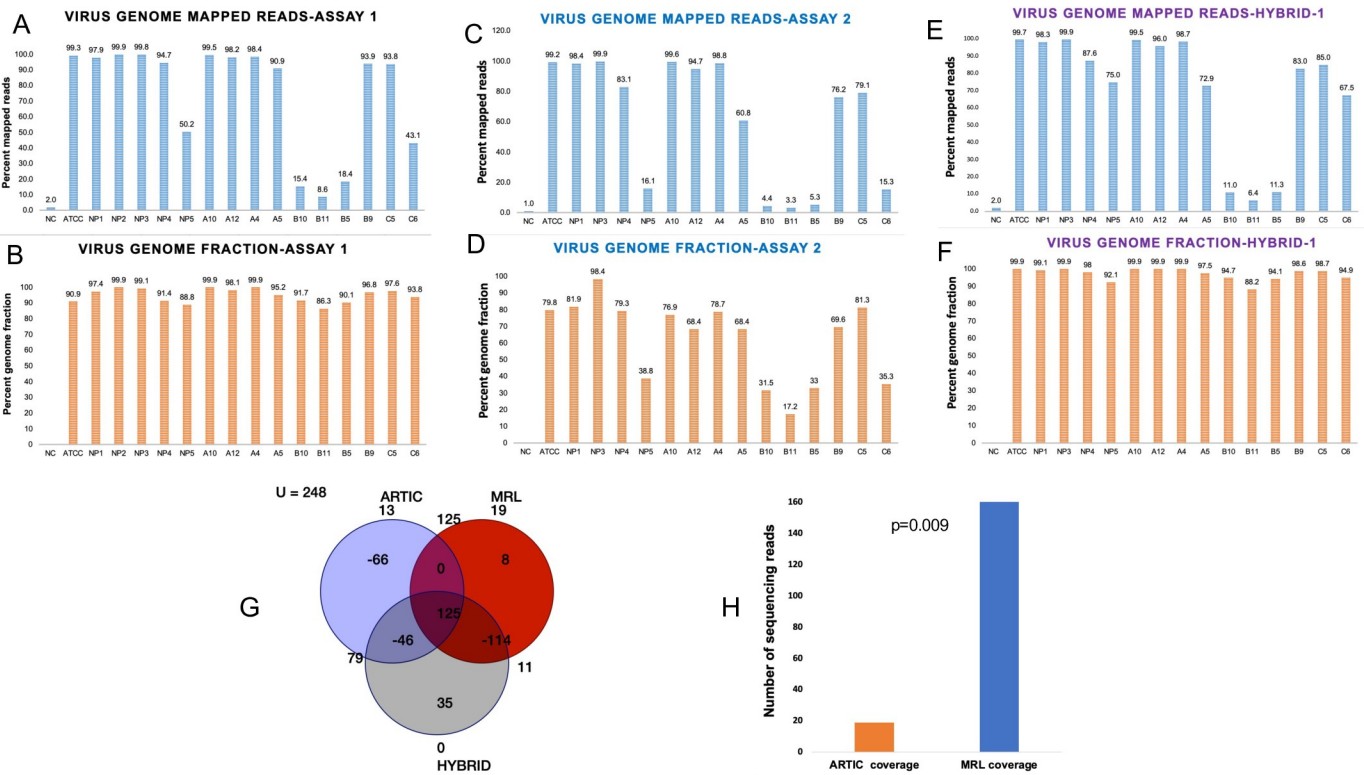

**Fig 2. Virus genome mapped reads and genomic fraction covered in various assays.** Percentage of virus genome mapped sequencing reads (A) and virus genome fraction covered with ARTIC primers (B). NC represents the negative control (Human Dendritic Cell RNA), PC represents the positive control (VR1986D-ATCC SARS-CoV-2 RNA), and fifteen nasopharyngeal swab RNAs from COVID-19 positive patients. Panels C and D, show the percentage of viral genome mapped sequencing reads (C) and virus genome fraction covered (D) with MRL primers. Data in panels E & F show the percentage of the viral genome mapped sequencing reads (E) and virus genome fraction covered (F) in ARTIC plus MRL Hybrid data set. The NP2 sample could not be analyzed with MRL primers due to limited sample material. Panel G: Venn-diagram summarizes the number of mutations that were uniquely or commonly detected by individual (ARTIC or MRL) primers and hybrid analysis. Panel G: Shows average read depth on ambiguous mutations in ARTIC assay. T-test p value for read depth comparison is shown.

loaded onto a MinION sequencer using a R9.4 flow cell. Sequencing was run for 48 hours. The FAST5 data was basecalled using ONT's Guppy pipeline.

## Data analysis

After trimming and removing the adapters, FASTQ files were mapped with bowtie2 [17] to the SARS-CoV-2 genome (NC_045512.2). To facilitate reproducibility, analysis samples were processed using the publicly available nf-core/viralrecon pipeline version 1.1.0 implemented in Nextflow 20.01.0 using Singularity 3.3.0 (10.5281/zenodo.3901628) [18–20]. Briefly, reads were trimmed using fastp [21], and de novo assembly was performed by spades, metaspades, unicycler, and minia. Variant calling was done by bcftools [22] using viral genome NC_045512.2, and variants were filtered where the BAQ was less than 20 or had a depth less than 10 reads. Minimum allele frequency for calling variants was set to 0.25, max was 0.75. De novo assembly was performed using spades, CoronaSPAdes, metaspades, unicycler and minia. Quast and Icarus were used to summarize contig and assembly statistics where the reference viral genome was NC_045512.2. FASTQ files from Illumina MiSeqDx runs were generated using bcl2fastq2. Sequencing reads were trimmed using Cutadapt (min Q30, adapter presence, shorter than 50 bases). The presence of host reads was detected using Kraken 2, the host genome version used was GRCh38 [23]. Fast5 files from Mk1C runs were base called and

demultiplexed with guppy (GPU enabled), PycoQC, FastQC and NanoPlot were used for sequencing QC visualizations [24]. The ARTIC network bioinformatics pipeline (https://github.com/artic-network/artic-ncov2019) was also used to generate a comprehensive view of our Oxford Nanopore long reads. Briefly, reads were mapped to NC_045512.2 with Minimap2 and BAM files were sorted and indexed [25]. Variants were called using Nanopolish where ploidy was set to 1 [26]. The minimum allele frequency was 0.15, the minimum flanking sequence was 10 bases, and at most, 1000000 haplotypes were considered. Combination of Illumina MiSeq paired end reads and Oxford Nanopore Mk1C long reads were assembled using SPAdes, hybrid assembly stats were summarized with Quast. Contiguous scaffolds were visualized with bandage (https://github.com/rrwick/Bandage). Pipeline automation was done by creating Nextflow workflows (v20.01.0).

## Results

### Assay-1: Short-amplicon sequencing results

We generated more than 500K sequencing reads on each study sample including the positive control ATCC SARS-CoV-2 RNA and fifteen COVID-19 positive patients that had their nasopharyngeal swabs tested by RT-PCR for the presence of SARS-CoV-2. All the patient samples had RT-PCR Ct values <30 (Table 1). The ARTIC primer generated >80% pass filter sequencing reads. The summary of sequencing metrics is given in Table 1. As shown, 99.26% of the sequencing reads on ATCC SARS-CoV-2 RNA mapped to SARS-CoV-2 reference genome (NC_045512.2). Similarly, over 90% of the sequencing reads in patient samples also mapped to the reference genome, except for a few samples (Table 1, Fig 2A). These viral reads in positive control and patient samples accounted for >94% of the virus genome (Table 1, Fig 2B). Variant analysis detected 209 single nucleotide variants (SNVs) and 8 deletions & insertions (S2 File). Of these 28 mutations were SARS-CoV-2 lineage defining variants including several variants of concern i.e. K417N, E484K, N501Y, P618H and variants of interest i.e. L18F, 69–70 Del, D80A, Y144Del, L242Del, A570D, A701V and T716I. Interestingly most of these new mutations were detected in a more recently collected samples. Variant analysis showed a range of mutations starting from minimum 14 and maximum 38 mutations per sample. Phylogenetic Assignment of Named Global Outbreak Lineages (PANGOLIN) lineage analysis on study samples identified four samples with B.1.1.7, three samples with B.1.2, two samples with B.1.617.1 and at least one sample with A, B.1.351, B.1.427, B.1.527, B.1.575.1 and P.1.1 lineages (Table 1). Interestingly, two specimens, A4 and B10 were assigned B.1.617.1 lineage that represent Kappa strain.

### Assay-2: Long-amplicon based sequencing results

Next, we analyzed the sequencing data generated with MRL primers: Assay-2. This assay did not include sample NP2 due to insufficient material. The sequencing metrics are described in Table. This assay generated 2–5 million sequencing reads on patient specimens. The percentage of reference mapped reads and genomic fractions are given in Table 1. The percentage of viral genome mapped reads and genomic fraction covered with MRL primers showed more variations among samples (Fig 2C and 2D. Variant analysis on assay-2 data identified a total of 156 mutations in study samples which included 152 SNVs and 4 insertion & deletion variants (S3 File). Of these 156 mutations, 125 (80%) were concordant with those called in the short-amplicon data, assay-1. The low coverage samples showed genotypic discordance in two data sets due to poor genomic coverage in either assay. Long-amplicon data captured 20 key lineage defining mutations including spike gene variants of concern K417N, E484K, N501Y and P618H (S3 File). Overall, long-amplicon assay detected 12 mutations that were either

ambiguous or not at all called in ARTIC assay (S4 File). Further in-depth analysis of the ambiguous base calls in assay-1 or assay-2 revealed that the observed discrepancy was either due to insufficient sequencing read depth or incorrect read alignments.

## Assay-1 & Assay-2 combined (Hybrid-1) analysis

To improve the sequencing coverage across poorly captured genomic segments in individual assays, we merged the sequencing reads from assay-1 and assay-2 to generate a hybrid assembly and then called the variants again. The hybrid data set has about 4–12 Million reads per sample (Table 1). As shown in Fig 2E and 2F, hybrid-I data improved both percentage of virus genome mapped reads as well as a fraction of covered genome (Fig 2E and 2F). This eventually improved variant detection overall and the resolution of lineage assignments, at least in one sample (NP1). Fig 2G summarizes unique and shared number of variants in individual and hybrid analyses. As shown, 125 variants were common in all the three analyses. Hybrid data identified a total of 218 mutations in study samples which included 210 SNVs and 8 insertion & deletion variants (S5 File). Investigation of mutations that were ambiguous in ARTIC data showed poor read quality and poor sequencing coverage across these 12 positions in the ARTIC data (Fig 2H).

Next, analysis of sequencing coverage in short (ARTIC) and long amplicon (MRL) data identified genomic intervals that were poorly captured in ARTIC assay (Fig 3). As illustrated

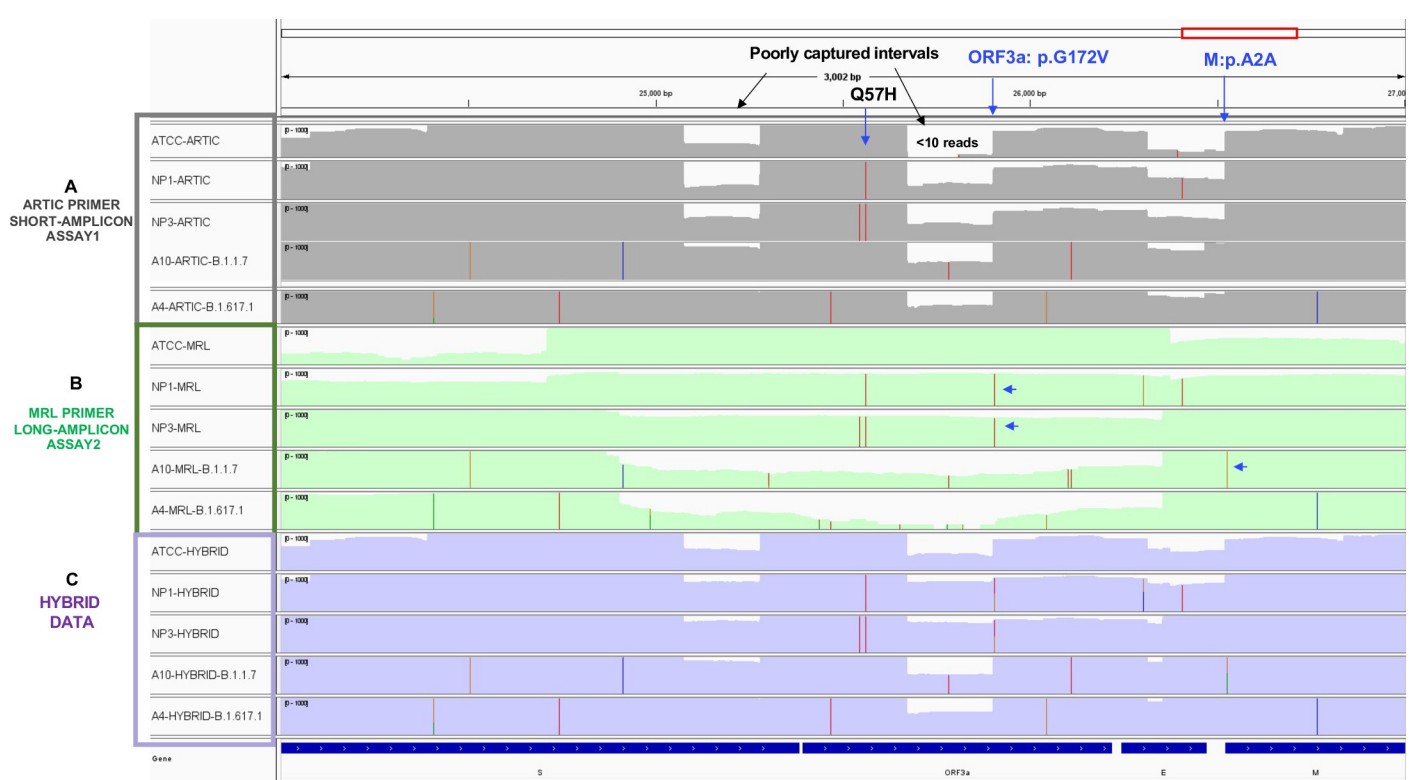

**Fig 3. Short plus long-amplicon hybrid data provide uniform and maximum genomic coverage.** The Integrative genomics viewer (IGV) plot shows sequencing coverage tracks in ATCC positive control and four patient samples based on ARTIC primers (short-amplicon data in panel A), MRL primers (long-amplicon in panel B) and Hybrid data (short + long-amplicons in panel C). Data is shown for ATCC RNA and four patient samples (NP1, NP3, A10 and A4) in each panel. The x-axis shows the genomic position in virus genome and the y-axis shows the individual samples. Top panel grey tracks represent ARTIC data, middle panel green tracks represent MRL data and bottom panel purple tracks represent Hybrid data. Colored lines on the sequencing coverage tracks indicate detected mutations. Black solid arrows point to the poorly captured genomic intervals in ARTIC data set. Blue solid arrows point to examples of three mutations, Q57H, G172V and A2A mutations.

in Fig 3A, short amplicon data in positive control ATCC RNA and four patient samples (NP1, NP3, A4, A10) shows genomic intervals that exhibit a sudden drop (<10 sequencing reads) in sequencing coverage across the Spike, ORF3a and E gene regions. On the other hand, long-amplicon sequencing on the exact same set of samples shows relatively uniform coverage across this region (Fig 3B green tracks). The colored lines on the coverage tracks indicate the detected mutations in patient samples. As shown, mutation, i.e.,Q57H in the genomic region that have good uniform sequencing coverage in both assays, was consistently picked up by both short (Panel 3A) as well as long-amplicon data (Panel 3B). However, a nonsynonymous mutation G172V in ORF3a gene at nt25907(G → T), and a synonymous mutation A2A in M gene at nt26528(A → G) located in the poorly captured region in ARTIC assay, were missed in short amplicon data (Panel A), whereas the long-amplicon data did detect G172V in samples NP1 and NP3, and A2A mutation in sample A10 (Panel B). Next, analysis of merged short and long-amplicon sequences confirmed this mutation with high confidence in Hybrid data as shown in Fig 3C. Overall hybrid data improved the detection of 12 mutations that were either ambiguous or not called in ARTIC data due to poor sequencing coverage or alignment issues (S4 File). Thus, hybrid data identified final set of 214 high confidence mutations that were included in-depth downstream phylogenetic analysis (S6 File).

## Lineage assignment and phylogenetic analysis on high confidence mutations

We assessed the mutation load in patient specimens that were collected several months apart. As shown in Fig 4A, first five samples (NP1-NP5, blue bars) were collected during the months of January-February 2021, whereas remaining ten samples (C6-A5, red bars) were collected during the months of May-June 2021. Consistent with literature that SARS-CoV-2 virus in gaining on an average 1–2 mutations per month, our data show increased number of mutations (~1.4 time) in May June specimens as compared to those collected in Jan-Feb 2021 (Fig 4A). It was further supported by the appearance of Alpha (B.1.1.7), Beta (B.1.351), Gamma (P.1.1) and Kappa (B.1.617.1) strains of virus in more recent samples (Fig 4A). The highest number of mutations in samples with B.1.617.1, B.1.1.7 and P.1.1. lineage of virus is consistent with literature documented trend of ongoing evolution in SARS-CoV-2 genome. Lineage assignment were mostly consistent between ARTIC only and Hybrid data except for NP1 sample which was assigned B.1.2 in ARTIC data, whereas Hybrid data assigned it B.1.596 lineage. Next, hybrid data analysis showed 8 mutations that were quite common (>40% frequency, blue label color) in study sample cohort (Fig 4B). These are shown with solid color bars in the Fig 4B. Amino acid changes in these common variants are indicated on the top of each bar. In addition, many CDC noted variants of concern and variants of interest were also observed to be accumulating in spike gene region (highlighted with red font color). This analysis also showed that several common and low frequency mutations are emerging in the *Spike*, *Membrane* and *Nucleocapsid* gene regions (Fig 4B).

There are endemic human coronaviruses HCoV-229E, NL63, OC43, and HKU1 that cause upper and lower respiratory tract infections in children and adults [27–29]. SARS-CoV-2 resembles these endemic viruses, original severe acute respiratory syndrome coronavirus (SARS-CoV-1) and Middle Eastern respiratory syndrome (MERS-CoV) [30]. So, to explore the evolutionary history and relationship among these different coronaviruses, we performed phylogenetic analysis on the study samples. A maximum likelihood phylogenetic tree was constructed on various endemic strains and present study specimens. As shown in Fig 5, all the endemic strains formed a tight clade that evolves into MERS, SARS1 and SARS-CoV-2 viruses. Except two study samples C6 and B11 that shared the clade with SARS and original SARS-CoV-2 strain (ATCC), all NP1-NP5 samples that represent SARS-CoV-2 virus in January-February 2021 formed one tight clade. On the other hand, most of the recently collected study

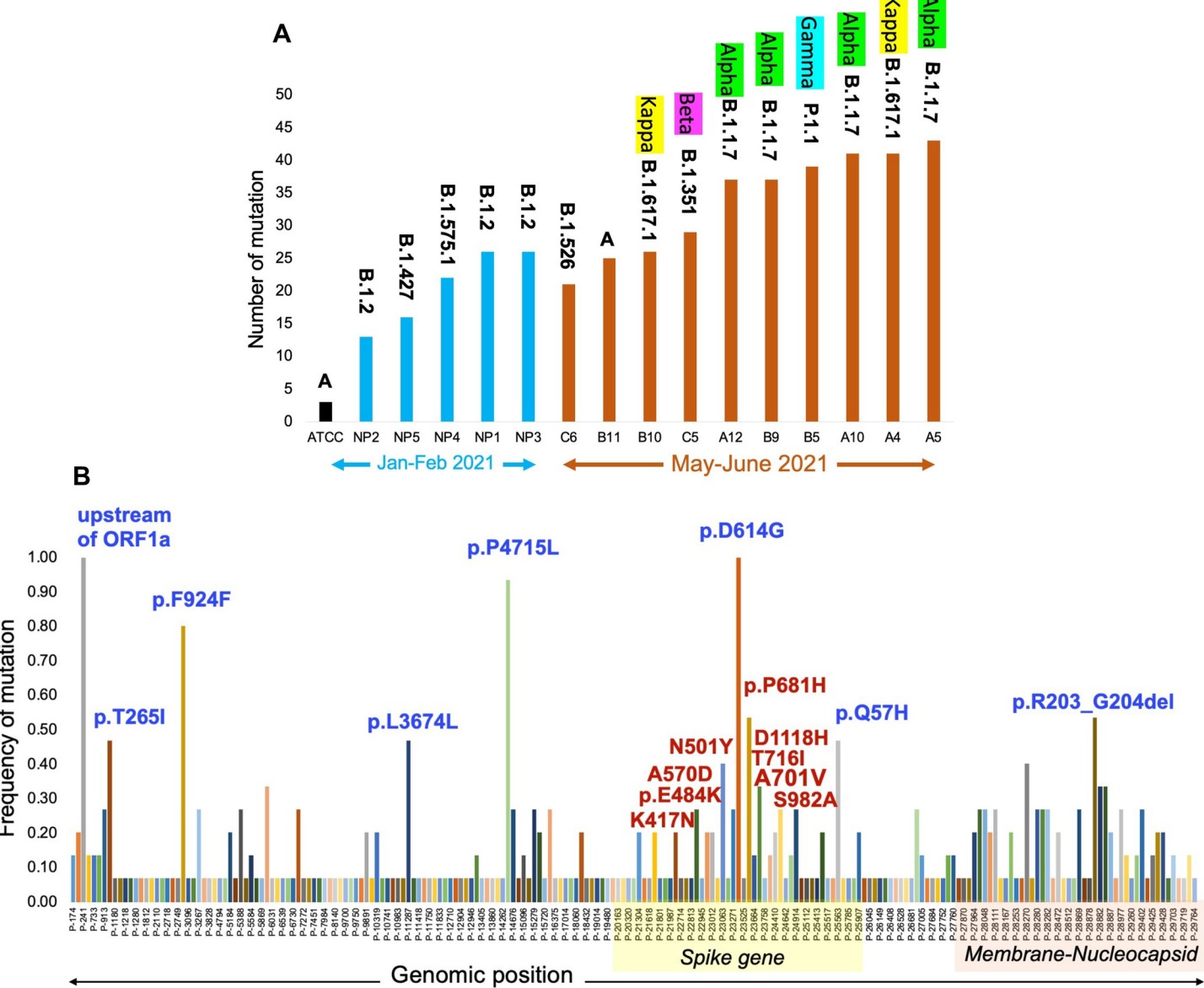

**Fig 4. Mutation load and pangolin lineage assignment in study samples.** Panel A shows number of mutations detected in ATCC RNA and fifteen patient samples in Hybrid data set. The x-axis shows the study samples. ATCC RNA is shown in black bar. NP1-NP5 samples shown as blue bars indicate samples collected during the month of Jan-Feb 2021. Samples C6-A5 were collected during the month of May-June 2021 are shown with red bars. The y-axis shows total number of mutations detected in each study sample. Pangolin lineage and WHO label for that lineage are shown on the top of individual bar. Panel B show frequency distribution of 214 high confidence mutations in the study samples. The x-axis shows genomic position of detected mutations. The y-axis shows the frequency of each mutation in the study sample. Each bar represents an individual mutation. In this study's cohort, 8 high bars show mutations with >40% frequency. Amino acid alterations are shown on the top of each bar for the 8 most common mutations (Blue font color) and several variants of concern and variants of interest in spike gene (Red font color).

specimens formed a distinct clade which showed significant genetic diversity and heterogeneity, suggesting ongoing viral evolution (Fig 5).

## Assay-3: MRL long-read sequencing analysis on MinION

To assess the performance of our MRL primers with the long-read sequencing pipeline from Oxford Nanopore Technology, we sequenced the long amplicons from the ATCC positive

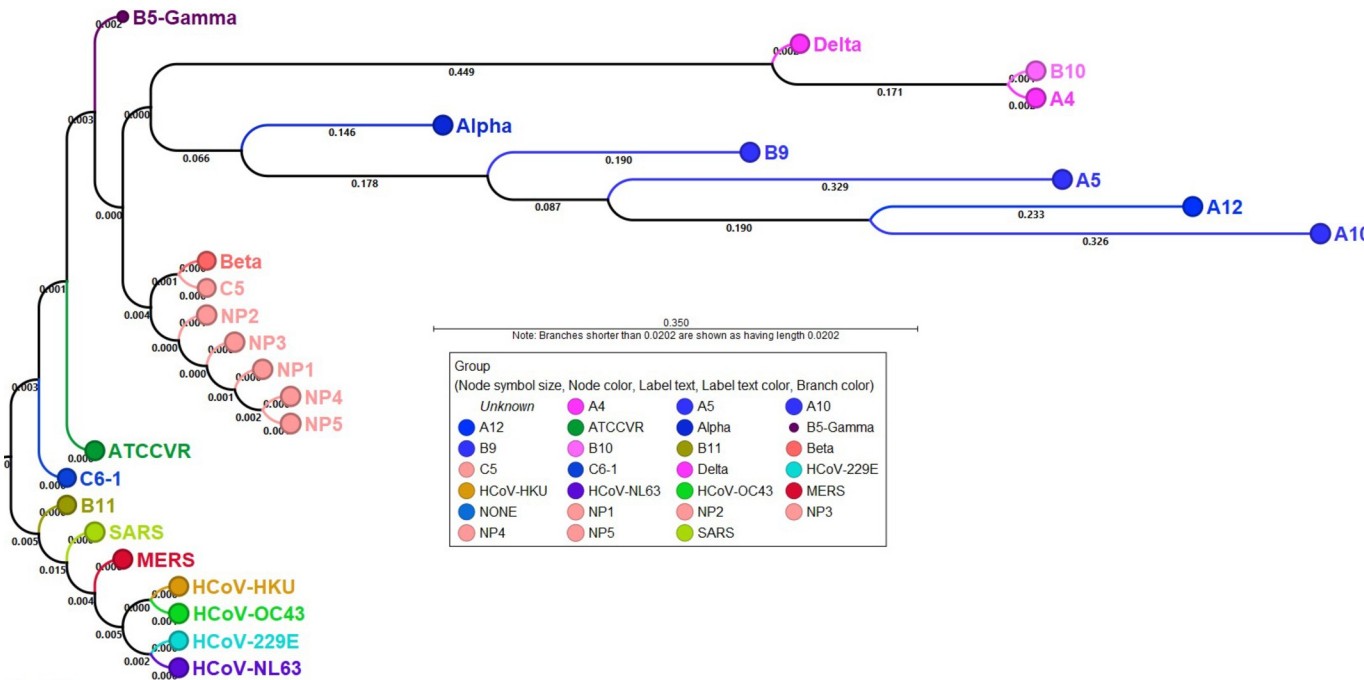

**Fig 5. Phylogenetic analysis on study specimens.** A maximum likelihood tree was constructed to explore the phylogenetic relationship between SARS-CoV-2, SARS-CoV-1, MERS and endemic coronaviruses (HCoV-NL63, HCoV-229E, HCoV-OC43, HCoV-HKU). The whole genome sequences for SARS, MERS and endemic coronaviruses were downloaded from NCBI. The numbers along the branches mark the bootstrap values percentage out of 1000 bootstrap resamplings. Samples NP1-NP5 that represent sampling from Jan-Feb 2021 form a clade and are highlighted with pink color. ATCCVR RNA shown in green font color and C6 and B11 patient sample are part of separate clade. Sample B9, A5, A12, and A10 shown in blue label font color cluster closely on tree. Dots labels as Alpha, Beta and Delta represent reference genomes for respective strains downloaded from NCBI.

control and three patient samples (NP1, NP3 & NP4) on MinION R.9.4.1 flow cell for 48 hours. These samples were selected based on the amount of available RNA sample. As summarized in S7A File, we generated 500K-1 million sequencing reads on each sample on MinION. More than 80% of the sequencing reads were mapped to the viral genome (Fig 6A). The genomic fraction covered was >95% in all four samples (Fig 6B, S7A File). We mapped the nanopore sequencing data to the reference genome and called the variants. In total, 77 mutations were detected in the nanopore sequencing data, of which 50 were concordant with ARTIC assay-1, and 48 were concordant with MRL assay-2 data from Illumina short-read sequencing (S6 File). Of 77 mutations, 72 were SNVs and 5 were insertion & deletions (S7B File). However, these insertions and deletions were only detected in nanopore data, so further validation and confirmation is needed. We also combined MRL long-read sequences with ARTIC short-read sequences as Hybrid-II data and observed improvement in the percent of reads mapping to the viral genome as well as in overall genomic coverage (Fig 6C and 6D). As shown in case of ATCC RNA, we observed that long-read sequencing data has more uniform sequencing coverage across the deletion-prone region of the spike gene than short-read ARTIC data. MRL long-read data shows uniform depth of coverage across the 2KB interval in the spike gene region as compared to short-amplicon data for ATCC positive control (Fig 6E and 6F). The bottom panel (Fig 6G) shows a snapshot of UCSC genome browser across the spike gene region where several micro and major deletions have been reported recently in emerging lineages of the SARS-CoV-2 virus (Fig 6G).

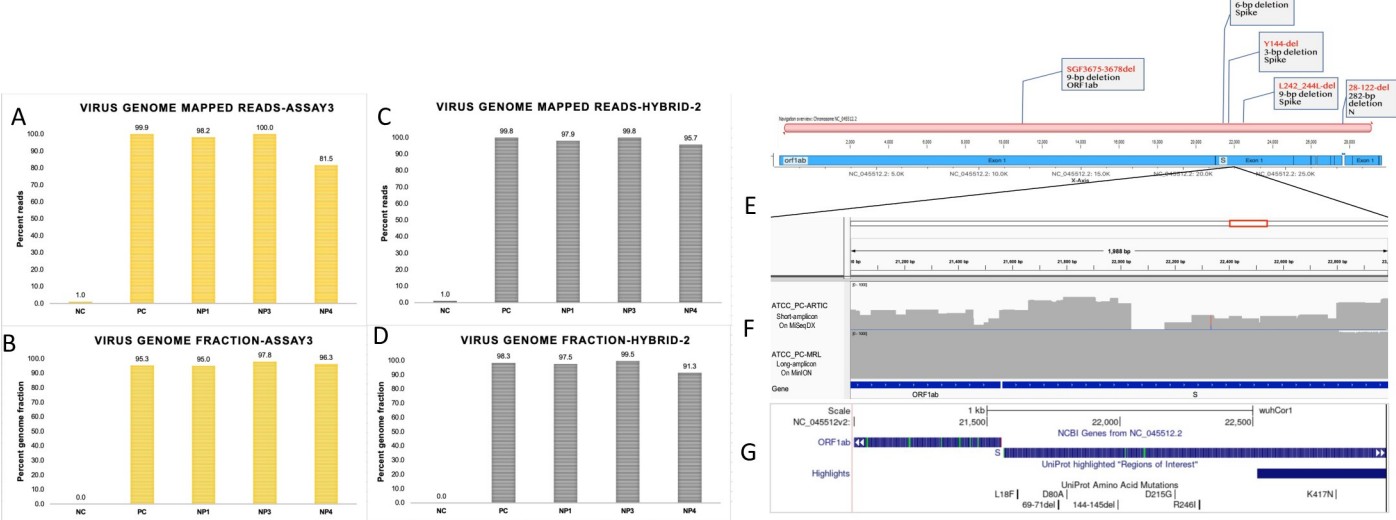

**Fig 6. Long-read sequencing provides uniform coverage across deletion-prone region in the virus genome.** Panel A shows the percentage of the virus genome mapped sequencing reads (A) and virus genome fraction covered with Nanopore sequencing data (B), in the positive control (VR1986D ATCC SARS-CoV-2 RNA). Panels C-D show reads mapped to the virus genome and covered genomic fraction in combined, ARTIC + Long-read (Hybrid-II) data, respectively. Panel E illustrates a gene sketch on known deletions in 2kb region of the spike gene. Panel F shows sequencing coverage across a deletion-prone region of the spike gene in ATCC positive control RNA in short-amplicon ARTIC and long-amplicon MRL data. Top coverage plot on ATCC_PC_ARTIC shows sequencing coverage in short-amplicon on Illumina MiSeqDx platform, whereas second track underneath show long-read sequencing data on ATCC_PC sample using MRL long-amplicon primers. Panel G shows a UCSC genome browser snapshot across spike gene region that show previously reported deletions in this genomic interval of SARS-CoV-2 genome.

## Discussion

High viral transmission in the ongoing COVID-19 pandemic is enabling SARS-CoV-2 to mutate at a faster rate, resulting in an abundance of new mutations in the genome [8, 31–34]. Since most of the currently used PCR primers, protocols, and sequencing strategies were developed on the original reference genome from the beginning of the pandemic, mutations in new strains of the virus might impact diagnostic and research methods [35]. The ARTIC primer pool that amplifies multiple short amplicons in a multiplex-PCR reaction is a widely used method to sequence the SARS-CoV-2 genome [16, 33, 36]. However, as no method is perfect, researchers have identified issues with existing short amplicon-based methods and provided potential alternative solutions [37–39]. Data from the present study (Fig 3) and others show that some key regions of the genome are poorly captured with ARTIC primers [40]. This could be due to poor performance of some primer pairs in multiplexed PCR due to either suboptimal PCR conditions within the assay or emerging mutations on primer binding sites [41]. Our approach to capture and assemble the genome using both short and long-amplicons improve the chance of capturing mutations in the intervals that may have uneven sequencing coverage on ARTIC assay. This strategy allows for variant calling with high confidence. Accurate detection of all the mutations in the virus genome is important for correct phylogenetic lineage assignment and functional studies. Although, overall lineage assignments in ARTIC and Hybrid data were the same, but we did observe improved resolution of lineage in case of NP1 sample which was assigned B.1.2 lineage in ARTIC assay and B.1.596 in Hybrid assay. We observed that on 8 of the 12 mutations that were not called in ARTIC assay had very poor sequencing depth i.e. <20 reads and the remaining 4 seems to have issues with read alignments that require further investigation. Similarly, the MRL assay did not picked 79 mutations that were detected in ARTIC assay. We speculate that it could be due to unsuccessful amplification

of some long amplicons in some samples due to poor Viral RNA complexity as a long RNA template is essential to generate long amplicons. Therefore, samples with degraded RNA may not yield good data with long amplicon PCR. This suggests that sequencing each sample with both short and long amplicon primers can provide more complete and comprehensive genomic coverage to accurately detect true mutations across the genome. The two primer set approach also allows confirmation of low frequency novel mutations and excludes sequencing and analysis related artifacts. We admit that sequencing with two sets of primers would increase the cost of reagents and processing time in the protocol. However, it is worth it for the accuracy and completeness of the genomic data, especially given that the virus is still mutating and several new mutations are currently emerging globally. Our assay can be employed to investigate clinically complicated specimens that require complete genomic coverage, such as those with large structural mutations. Although our findings are based on a small number of samples, but we have demonstrated a strategy to capture the mutations in SARS-CoV-2 genome more accurately and efficiently.

## Supporting information

**S1 File. MRL long-amplicon primer sequences.**
(XLSX)

**S2 File. MRL long-amplicon primer sequences: List of variants called in Assay-1 (ARTIC).**
(XLSX)

**S3 File. List of variants called in Assay-2 (MRL).**
(XLSX)

**S4 File. List of eight mutations with ambiguous or no call in ARTIC data.**
(XLSX)

**S5 File. List of variants called in hybrid data set.**
(XLSX)

**S6 File. Final set of high confidence variants in hybrid assay used for downstream analysis.**
(XLSX)

**S7 File. Summary of sequencing metrics on MinION sequencing.**
(XLSX)

## Acknowledgments

Authors gratefully acknowledge donors of de-identified patients who provided nasopharyngeal swabs samples for this study.

The authors acknowledge the Texas Advanced Computing Center (TACC) at The University of Texas at Austin for providing HPC and visualization resources that have contributed to the research results reported within this paper. URL: http://www.tacc.utexas.edu. Our sincere thanks to Kaylee D for her diligent proofreading of the manuscript.

## Author Contributions

**Conceptualization:** Prithvi Raj.

**Data curation:** Carlos Arana, Matthew Brock, Brandi Cantarel, Jeffrey SoRelle.

**Formal analysis:** Carlos Arana, Matthew Brock, Li Chen.

**Investigation:** Chaoying Liang, Prithvi Raj.

**Methodology:** Carlos Arana, Chaoying Liang, Matthew Brock, Bo Zhang, Li Chen, Prithvi Raj.

**Project administration:** Jinchun Zhou.

**Resources:** Bo Zhang, Jeffrey SoRelle, Lora V. Hooper, Prithvi Raj.

**Software:** Carlos Arana, Brandi Cantarel.

**Supervision:** Prithvi Raj.

**Validation:** Jinchun Zhou, Prithvi Raj.

**Visualization:** Jeffrey SoRelle.

**Writing – original draft:** Jeffrey SoRelle, Lora V. Hooper, Prithvi Raj.

**Writing – review & editing:** Lora V. Hooper, Prithvi Raj.

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
