## [Decision Letter · Decision Letter 0]

26 Jul 2021

PONE-D-21-19442

A Short Plus Long-Amplicon Based Sequencing Approach Improves Genomic Coverage and Variant Detection In the SARS-CoV-2 Genome

PLOS ONE

Dear Dr. Raj,

Thank you for submitting your manuscript to PLOS ONE. After careful consideration, we feel that it has merit but does not fully meet PLOS ONE’s publication criteria as it currently stands. Therefore, we invite you to submit a revised version of the manuscript that addresses the points raised during the review process.

We look forward to receiving your revised manuscript.

Kind regards,

Chandrabose Selvaraj, Ph.D.

Academic Editor

PLOS ONE

2. We note that your ethics statement in the online submission form states that the ethics committee granted approval, whilst the methods state that a waiver was granted. Please ensure you have accurately reported whether approval or a waiver was granted in both places. In addition, please state whether samples were deidentified before you had access to them.

“NO”

“NONE”.

6. Please include captions for your Supporting Information files at the end of your manuscript, and update any in-text citations to match accordingly. Please see our Supporting Information guidelines for more information: http://journals.plos.org/plosone/s/supporting-information

Additional Editor Comments (if provided):

1. Kindly ensure to provide the proper updated references in appropriate places

Reviewers' comments:

Reviewer's Responses to Questions

5. Review Comments to the Author

Reviewer #1: Article entitled “A Short Plus Long-Amplicon Based Sequencing Approach Improves Genomic Coverage and Variant Detection In the SARS-CoV-2 Genome” provides an approach to supplement ARTIC’s short amplicon sequences with long amplicons to generate more complete and high-quality sequencing data for mutation detection and phylogenetic analysis. This article may be considered after revision.

Comments

• Nasopharyngeal swabs were taken only from 5 patients , is that sufficient for the analysis?

• Here, They reported two of the spike gene variants at nucleotide positions 23604 (P681H) & 23709 (T716I) , ? is it novel variants?

• Main claim is new strategy that capture the mutating SARS-CoV-2 genome more accurately and efficiently, allowing for improved analysis of genomic information on evolving strains. How this approach is better when compare to other approaches ?

• Is these any similar approach which reported with better efficiency?

• Results parts need more discussion comparing other variants linked to specific population.

• Are the five COVID-19 positive patients from same location ?

Reviewer #2: "A Short Plus Long-Amplicon Based Sequencing Approach Improves Genomic Coverage and Variant Detection In the SARS-CoV-2 Genome" by Arana et al. describes a study comparing short amplicons, long amplicons, and a hybrid analysis approach for tiled amplicon genome sequencing of SARS-CoV-2. The study design and analysis is systematic and allows for robust comparison between the methods. The ability to minimize drop outs due to deletions or under-amplification with a given short amplicon primer pair is an advantage of this method. Several questions and requested clarifications are below, but pending these revsions I support the publication of this manuscript in PLOS One.

Major points and questions

-We routinely get high (>99.5%) coverage with ARTIC primers, though this is dependent on the sequencing depth of the sample and also sensitive to Ct and sample type. Do you have Ct values or other estimates of viral load for the samples that were analyzed in this study?

-Figure 2B,D,F - Is this coverage at at least 1x or with some other threshold applied?

-Figure 2B,D,F - The ARTIC primers don't cover 100 percent of the viral genome (and you will also lose some coverage at the 5' and 3' ends if primers are trimmed), so it is surprising that some of these bars say 100%. I am guessing this is rounded from 99 point something, but the authors should include another decimal place or otherwise clarify these coverage numbers.

-Figure 2G - Are these variants called with a minimum read depth threshold as indicated in the methods? Which of the three variant calling pipelines was used to generate these numbers (or was it a consensus between the three tools)?

-Figure 2G - It might be good to include a similar Venn diagram (as panel H or a supplement) that summarizes the overlap if only regions with adequate coverage in both data sets are considered to complement the discussion at the beginning of page 15.

-Figure 2G - The Venn diagram seems to indicate 60 mutations present in more than one analysis (41 + 10 + 4 + 5), but the text and Table 6 lists 59. What is behind this discrepancy?

-Figure 4B - I am not sure that the MJ network figure provides useful information since these are just 5 random patient samples. It might be better to show the Nextstrain plot with these samples highlighted to put them in context of the global viral diversity.

-Discussion - Given that high coverage can be routinely obtained with short amplicons in most cases and the fact that adding a second library prep and sequencing dataset adds significant cost and time to the analysis workflow (the "slightly increase" wording is deceptive and should be changed as this at least doubles the cost). It would be useful to reframe the discussion to focus on when application of this technique would be most beneficial. I think it is hard to imagine (and likely overkill) for most routine large-scale surveillance sequencing, but may be useful as a reflex option for problematic samples or in a clinical context where complete coverage and additional confidence in the data is required.

Minor points

-Page 8 - Abstract: Please qualify first sentence "High viral transmission in the COVID-19 pandemic has enabled SARS‐ CoV‐ 2 to acquire new mutations that *may* impact genome sequencing methods."

-Page 9 - Introduction: "overtime" should be two words "over time."

-Page 12 - Step 4: Sequencing section. Please define "quality pass sequencing reads" or clarify this phrase.

-Page 12-13. The data analysis seems to be described twice (once in the sequencing section and once in the data analysis section). Please consolidate these descriptions.

Reviewer #3: In this paper, authors have used three methods including long and short read technologies for sequencing of SARS-CoV-2 genome and concluded that, combination of short and long sequencing reads gives better representation of SARS-CoV-2 variants. However, I have few concern mentioned below.

1. Nowadays, scientific community is looking for cheaper alternatives and wanted to reduce the cost of sequencing as much as low. In this direction, authors have used two different sequencing technologies to achieve the target, which in turn, increase the cost per sample.

2. Authors have used only five samples for sequencing. Were all the samples of the same lineage? or of different lineages? If, they were of the same lineage, authors must validate their long-amplicon primers on different lineages of the SARS-CoV-2.

3. Why authors have checked the quality of RNA on bioanalyzer? Which module, I mean Prokaryotic or Eukaryotic was used to run? Based on bioanalyzer results, how the quality of viral RNA was decided?

4. The Thermo Fisher Scientific, SARS-CoV-2 NGS panel consist of two pools with amplicons ranging from 125–275bp in length for complete coverage of over 99% of the viral genome and variants and that is irrespective of the virial lineage. How your method is superior to this?

---

## [Author Response · Author response to Decision Letter 0]

1 Oct 2021

PONE-D-21-19442 

A Short Plus Long-Amplicon Based Sequencing Approach Improves Genomic Coverage and Variant Detection In the SARS-CoV-2 Genome

Authors thank reviewers and Editors for taking time to read our manuscript and provide valuable suggestions to further improve it. We have responded to each and every point raised by the reviewers. We have added more samples and performed additional data analysis. All the new data have been included in the revised manuscript.

Reviewer #1: Article entitled “A Short Plus Long-Amplicon Based Sequencing Approach Improves Genomic Coverage and Variant Detection In the SARS-CoV-2 Genome” provides an approach to supplement ARTIC’s short amplicon sequences with long amplicons to generate more complete and high-quality sequencing data for mutation detection and phylogenetic analysis. This article may be considered after revision.

Comments

• Nasopharyngeal swabs were taken only from 5 patients , is that sufficient for the analysis?

Ans: More samples are always good. But to assess the genomic coverage and variant detection consistency of two types of assays, we believe five samples were sufficient. Anyway, we have sequenced 10 additional specimens, including alpha, beta and gamma strains of virus to increase the sample size in the revised manuscript.

• Here, They reported two of the spike gene variants at nucleotide positions 23604 (P681H) & 23709 (T716I) , ? is it novel variants?

Ans: These two spike variants are part of B1.1.7 lineage and were relatively less common but recently emerged mutations. C23604A variant is a nonsynonymous mutation at exonic region of S: p.P681H. Frequency of this mutation is highest in the specimens analyzed in United States. On the other hand, C23709T change is a nonsynonymous mutation at exonic region of Spike p.T716I. This is relatively less common variant overall. But it was interesting to find these two mutations in 1 out of 5 specimens. Our new data analysis shown several new variants in addition to these two.

• Main claim is new strategy that capture the mutating SARS-CoV-2 genome more accurately and efficiently, allowing for improved analysis of genomic information on evolving strains. How this approach is better when compare to other approaches?

Ans: Large number of SARS-CoV-2 genome sequencing studies have used ARTIC primers. ARTIC design amplifies hundreds of short amplicons to assemble the full viral genome. This method was successfully used throughout the pandemic. These primers were designed based on the original, Wuhan strain of the virus. But, with ongoing pandemic, we know that virus has mutated significantly as evidenced by the emergence of several new and more transmissible strains i.e. B.1.1.7, B.1.35, B.1.617, P.1, B.1.617.1, etc. Therefore, we hypothesize that increasing mutation load in evolving virus may alter the original primer binding sites resulting in primer drop out and loss of data. Our approach amplify each specimen using both, short amplicon primers (ARTIC) as well as long-amplicon primers (MRL) and bioinformatically merge two data sets to assemble complete genome. Long amplicons and long-read sequencing has two major advantages. First: Using less number of primers to amplify genome reduces probability of encountering a mutated site. Second, long amplicons sequenced with long-read sequencing technology allow more accurate capture of structural variants such as large deletions and insertions. So, with these merits in our method, we strongly believe that a hybrid data analysis using both ARTIC plus MRL primers can improve the data quality, especially in the genomic intervals that are poorly captured by individual primers.

• Is these any similar approach which reported with better efficiency?

Ans: ARTIC primers and protocols have been extensively and widely used. Our work can further improve the existing method, especially in cases with higher mutational load and novel deletions and insertions.

• Results parts need more discussion comparing other variants linked to specific population.

Ans: We have added more discussion on population specific variants, as suggested. 

• Are the five COVID-19 positive patients from same location?

Ans: Yes. These specimens were collected and analyzed at UT Southwestern Medical Center.

Reviewer #2: "A Short Plus Long-Amplicon Based Sequencing Approach Improves Genomic Coverage and Variant Detection In the SARS-CoV-2 Genome" by Arana et al. describes a study comparing short amplicons, long amplicons, and a hybrid analysis approach for tiled amplicon genome sequencing of SARS-CoV-2. The study design and analysis is systematic and allows for robust comparison between the methods. The ability to minimize drop outs due to deletions or under-amplification with a given short amplicon primer pair is an advantage of this method. Several questions and requested clarifications are below, but pending these revsions I support the publication of this manuscript in PLOS One.

Major points and questions

We routinely get high (>99.5%) coverage with ARTIC primers, though this is dependent on the sequencing depth of the sample and also sensitive to Ct and sample type. Do you have Ct values or other estimates of viral load for the samples that were analyzed in this study?

Ans: I agree, ARTIC primers provides high coverage in general, but we did perform additional sequencing on couple of specimens, that did not improve the unevenness throughout. There are certain intervals that are still poorly captured even after additional sequencing. In the revised manuscript, we have now analyzed 10 additional samples, that do have precise Ct values see Table2. We do observe some association between CT value and covered genomic fraction in the assay. As shown in Table 2, A10 sample has lowest Ct value (5.4) and highest (99.90%) genomic coverage. On the other hand, B11 sample has Ct value of 25 and genomic fraction covered is 86.30%. Since specimens were de-identified, we could only get many other clinical details on these samples. I agree, heterogeneity in coverage could be attributed to original viral load.

-Figure 2B,D,F - Is this coverage at least 1x or with some other threshold applied?

Ans: We have provided coverage at both 1x and 10x in the revised manuscript.

-Figure 2B,D,F - The ARTIC primers don't cover 100 percent of the viral genome (and you will also lose some coverage at the 5' and 3' ends if primers are trimmed), so it is surprising that some of these bars say 100%. I am guessing this is rounded from 99 point something, but the authors should include another decimal place or otherwise clarify these coverage numbers.

Ans: Yes, these numbers are rounded. We have updated these plots with exact decimal points in the revised manuscript. 

-Figure 2G - Are these variants called with a minimum read depth threshold as indicated in the methods? Which of the three variant calling pipelines was used to generate these numbers (or was it a consensus between the three tools)?

Ans: Now with additional samples analyzed, mutations that were called in Hybrid data set were used for the downstream analysis. Venn diagram presents variants shared in various analyses in new data set. 

-Figure 2G - It might be good to include a similar Venn diagram (as panel H or a supplement) that summarizes the overlap if only regions with adequate coverage in both data sets are considered to complement the discussion at the beginning of page 15.

Ans: Instead of a Venn diagram, we have shown that discordant mutations have significantly lower sequencing coverage in ARTIC assay as compared to MRL assay (Fig. 2H). Similar was true in case of mutations that were captured by ARTIC assay but missed by MRL. Again, this suggest, hybrid data analysis is still a reasonable approach to improve the coverage and variant detection, especially in poorly captured genomic intervals in either assay.

-Figure 2G - The Venn diagram seems to indicate 60 mutations present in more than one analysis (41 + 10 + 4 + 5), but the text and Table 6 lists 59. What is behind this discrepancy?

Ans: Thanks for catching this. The final number was actually 60 in the old data. But since now we have sequenced and analyzed 10 more samples in the revised manuscript, the final number of high confidence mutations is 214 (Table 7). These were identified based on their call in Hybrid data set. Details on these final set of mutations are provided in the Table 7 of revised manuscript. 

-Figure 4B - I am not sure that the MJ network figure provides useful information since these are just 5 random patient samples. It might be better to show the Nextstrain plot with these samples highlighted to put them in context of the global viral diversity.

Ans: As per Reviewer’s suggestions, we have excluded MJ network from the revised manuscript. Instead, we have provided a maximum likelihood phylogenetic analysis and a Nextstrain plot on study samples.

-Discussion - Given that high coverage can be routinely obtained with short amplicons in most cases and the fact that adding a second library prep and sequencing dataset adds significant cost and time to the analysis workflow (the "slightly increase" wording is deceptive and should be changed as this at least doubles the cost). It would be useful to reframe the discussion to focus on when application of this technique would be most beneficial. I think it is hard to imagine (and likely overkill) for most routine large-scale surveillance sequencing, but may be useful as a reflex option for problematic samples or in a clinical context where complete coverage and additional confidence in the data is required.

Ans: We have reframed our statements as suggested and updated the discussion section in the revised manuscript. 

Minor points

-Page 8 - Abstract: Please qualify first sentence "High viral transmission in the COVID-19 pandemic has enabled SARS‐ CoV‐2 to acquire new mutations that *may* impact genome sequencing methods."

Ans: This correction has been made in the revised manuscript.

-Page 9 - Introduction: "overtime" should be two words "over time."

Ans: This correction has been made in the revised manuscript

-Page 12 - Step 4: Sequencing section. Please define "quality pass sequencing reads" or clarify this phrase.

Ans: Sequencing reads QC means that reads were trimmed using Cutadapt (min Q30, adapter presence, shorter than 50 bases). The presence of host reads was detected using Kraken 2. This section is updated and more information is provided in the revised manuscript.

-Page 12-13. The data analysis seems to be described twice (once in the sequencing section and once in the data analysis section). Please consolidate these descriptions.

Ans: This is corrected in the revised manuscript

Reviewer #3: In this paper, authors have used three methods including long and short read technologies for sequencing of SARS-CoV-2 genome and concluded that, combination of short and long sequencing reads gives better representation of SARS-CoV-2 variants. However, I have few concern mentioned below.

1. Nowadays, scientific community is looking for cheaper alternatives and wanted to reduce the cost of sequencing as much as low. In this direction, authors have used two different sequencing technologies to achieve the target, which in turn, increase the cost per sample.

Ans: I agree. Our method uses two assays instead of one, which certainly increases the cost per sample but the advantage is more complete data, which is important too. Due to cost, our method may not fit for very routine virus surveillance but certainly useful to profile clinically complex specimens that require more complete genomic coverage to profile novel structural variants i.e. deletions and insertions. 

2. Authors have used only five samples for sequencing. Were all the samples of the same lineage? or of different lineages? If, they were of the same lineage, authors must validate their long-amplicon primers on different lineages of the SARS-CoV-2.

Ans: These were from different lineages but not huge diversity. Now we have sequenced additional strains to further test and validate our primers and sequencing pipeline, as suggested by reviewer. Data on new samples with Alpha, Beta and Gamma strains of virus have been included in the revised manuscript.

3. Why authors have checked the quality of RNA on bioanalyzer? Which module, I mean Prokaryotic or Eukaryotic was used to run? Based on bioanalyzer results, how the quality of viral RNA was decided?

Ans: Sorry for the confusion, we have corrected this. Given the limited amount of RNA on these specimens, we have not checked quality/quantity on input RNA. RT-PCR Ct value of<30 was considered as inclusion criterion. We have used Tapestation 5000 to do the quality control on cDNA and final sequencing libraries. 

4. The Thermo Fisher Scientific, SARS-CoV-2 NGS panel consist of two pools with amplicons ranging from 125–275bp in length for complete coverage of over 99% of the viral genome and variants and that is irrespective of the virial lineage. How your method is superior to this?

Ans: Most of the currently employed methods use pools of primers that amplify multiple overlapping amplicons to assemble the viral genome. We noticed that some primer pairs (in case of ARTIC) have very low sequencing coverage, which results in loss of genomic information. To improve the variant detection across such low coverage intervals, our long amplicons provides uniform sequencing coverage enabling variant detection and accurate genotyping of the virus. In summary, our method does not replace existing methods but complement them to generate better genomic data, especially across the poorly captured regions with short amplicons. Our new data shows that our approach works well with emerging strains as well.

---

## [Decision Letter · Decision Letter 1]

1 Nov 2021

PONE-D-21-19442R1A Short Plus Long-Amplicon Based Sequencing Approach Improves Genomic Coverage and Variant Detection In the SARS-CoV-2 GenomePLOS ONE

Dear Dr. Raj,

Thank you for submitting your manuscript to PLOS ONE. After careful consideration, we feel that it has merit but does not fully meet PLOS ONE’s publication criteria as it currently stands. Therefore, we invite you to submit a revised version of the manuscript that addresses the points raised during the review process.

We look forward to receiving your revised manuscript.

Kind regards,

Chandrabose Selvaraj, Ph.D.

Academic Editor

PLOS ONE

Journal Requirements:

Reviewers' comments:

Reviewer's Responses to Questions

6. Review Comments to the Author

Reviewer #1: all comments have been addressed in the revised version , this article may be accepted.

Reviewer #2: The revised manuscript, "A Short Plus Long-Amplicon Based Sequencing Approach Improves Genomic Coverage and Variant Detection In the SARS-CoV-2 Genome" by Arana et al. has satisfactorily addressed all of my major comments and concerns. Aside from a couple of minor suggestions to improve the clarity of the figures (below), I think the data and interpretations are solid and support publication.

Minor points

-Figure 2: Harmonizing the labels (Assay I, Assay II, etc.) with those used in Figure 1 would help to make the figure more easily interpretable. Numbers are used for the assays in Figure 1 and 3, but roman numerals are used in Figure 2.

-Figure 6: Coverage values are rounded to the nearest integer. I suggest adding an additional decimal point as in Figure 2.

---

## [Author Response · Author response to Decision Letter 1]

18 Nov 2021

We have updated the figure 2 and figure 6 as per reviewer's suggestion. Revised manuscript has updated figures.

---

## [Editor Report · Decision Letter 2]

23 Nov 2021

A Short Plus Long-Amplicon Based Sequencing Approach Improves Genomic Coverage and Variant Detection In the SARS-CoV-2 Genome

PONE-D-21-19442R2

Dear Dr. Raj,

We’re pleased to inform you that your manuscript has been judged scientifically suitable for publication and will be formally accepted for publication once it meets all outstanding technical requirements.

Kind regards,

Chandrabose Selvaraj, Ph.D.

Academic Editor

PLOS ONE

---

## [Editor Report · Acceptance letter]

28 Dec 2021

PONE-D-21-19442R2 

A Short Plus Long-Amplicon Based Sequencing Approach Improves Genomic Coverage and Variant Detection In the SARS-CoV-2 Genome 

Dear Dr. Raj:

I'm pleased to inform you that your manuscript has been deemed suitable for publication in PLOS ONE. Congratulations! Your manuscript is now with our production department. 

Kind regards, 

on behalf of

Dr. Chandrabose Selvaraj 

Academic Editor

PLOS ONE